 

# Whole genome-based genetic insights of *bla*NDM producing clinical *E. coli* isolates in hospital settings of Pakistant

Sabahat Abdullah,[1] Abdulrahman Almusallam,[2] Mei Li,[2] Muhammad Shahid Mahmood,[1] Muhammad Ahmad Mushtaq,[1] Nahla O. Eltai,[3] Mark A. Toleman,[2] Mashkoor Mohsin[1]

**ABSTRACT**  Carbapenem resistance among Enterobacterales has become a global health concern. Clinical *Escherichia coli* isolates producing the metallo β-lactamase NDM have been isolated from two hospitals in Faisalabad, Pakistan. These *E. coli* strains were characterized by MALDI-TOF, PCR, antimicrobial susceptibility testing, XbaI and S1 nuclease pulsed-field gel electrophoresis (PFGE), conjugation assay, DNA hybridization, whole genome sequencing, bioinformatic analysis, and *Galleria mellonella* experiments. Thirty-four *bla*NDM producing *E. coli* strains were identified among 52 nonduplicate carbapenem-resistant strains. More than 90% of the isolates were found to be multidrug resistant by antimicrobial susceptibility testing. S1 PFGE confirmed the presence of *bla*NDM gene on plasmids ranging from 40 kbps to 250 kbps, and conjugation assays demonstrated transfer frequencies of *bla*NDM harboring plasmids ranging from $1.59 \times 10^{-1}$ to $6.46 \times 10^{-8}$ per donor. Whole genome sequencing analysis revealed *bla*NDM-5 as the prominent NDM subtype with the highest prevalence of *bla*OXA-1, *bla*CTX-M-15, *aadA2*, *aac(6')-Ib-cr*, and *tet(A)* associated resistant determinants. *E. coli* sequence types: ST405, ST361, and ST167 were prominent, and plasmid Inc types: FII, FIA, FIB, FIC, X3, R, and Y, were observed among all isolates. The genetic environment of *bla*NDM region on IncF plasmids included partial IS*Aba125*, the bleomycin *ble* gene, and a class I integron. The virulence genes *terC, traT, gad, fyuA, irp2, capU,* and *sitA* were frequently observed, and *G. mellonella* experiments showed that virulence correlated with the number of virulence determinants. A strong infection control management in the hospital is necessary to check the emergence of carbapenem resistance in Gram-negative bacteria.

**IMPORTANCE**  We describe a detailed analysis of highly resistant clinical *E. coli* isolates from two tertiary care centers in Pakistan including carbapenem resistance as well as common co-resistance mechanisms. South Asia has a huge problem with highly resistant *E. coli*. However, we find that though these isolates are very difficult to treat they are of low virulence. Thus the Western world has an increasing problem with virulent *E. coli* that are mostly of low antibiotic resistance, whereas, South Asia has an increasing problem with highly resistant *E. coli* that are of low virulence potential. These observations allow us to start to devise methodologies to limit both virulence and resistance and combat problems in developing nations as well as the Western world.

**KEYWORDS**  *E. coli*, *Galleria mellonella*, *bla*NDM, ST167, carbapenemase

Address correspondence to Mark A. Toleman, tolemanma@cardiff.ac.uk, or Mashkoor Mohsin, mashkoormohsin@uaf.edu.pk.

The authors declare no conflict of interest.

See the funding table on p. 14.

Antimicrobial resistance (AMR), particularly in Gram-negative bacteria, is one of the greatest global challenges to public health systems (1). Morbidity and mortality caused by multidrug-resistant (MDR) bacteria are increasing globally, with a recent study estimating the global burden of AMR at 4.95 million deaths in 2019 (2). Carbapenems have been recognized as last-resort antibiotics due to their broad spectrum

of antibacterial activity for severe infectious diseases caused by multidrug-resistant bacteria (3). However, their increased clinical use leads to the development of carbapenem-resistant Enterobacterales (CRE), responsible for healthcare-associated infections. CRE do not respond to commonly available antibiotics and are frequently associated with high mortality (4). In Asia, the resistance rate of Enterobacterales to imipenem and meropenem rose from 0.8%–1.2% and 1.0% to 1.3%, respectively, from 2001 to 2012 (5). In Pakistan, carbapenem resistance of *E. coli* and *Klebsiella pneumoniae* had risen from 1%–5% and 3%–18%, respectively, from 2009 to 2014 (6). Different carbapenemases produced by these bacteria inactivate carbapenems; NDM is the main carbapenemase found throughout South Asia and is the most clinically significant because of its rapid and ongoing evolution and global dissemination (7).

Since a *K. pneumoniae* NDM-bearing strain was reported first time from a Swedish patient who had received prior treatment in New Delhi, NDM has spread worldwide with common links to South Asia (8),—48 variants of NDM (NDM-1–48) have been detected so far (9). NDM-1, 4, 5, 6, and 7 are most prevalent worldwide, and NDM-5 is the most prevalent in South Asia and China (10). In contrast, several other types of carbapenemases, such as KPC, OXA-48, IMP, and VIM, have been found to be more common in other countries (11, 12). The highest distribution of NDM-positive species is observed in *K. pneumoniae* and *E. coli* (13). Notably, one of the primary reasons for the rapid emergence and spread of NDM is its close association with *E. coli* carried by the vast majority of humans, in addition to its close association with different mobile genetic elements such as insertion sequences, ISCR elements, plasmids, other transposons, and integrons. Conjugative plasmids such as Incompatibility groups (Inc) F, L/M, N, A/C, and X are commonly associated with the spread of $bla_{NDM}$ via horizontal gene transfer (HGT) (14).

The Indian subcontinent is the most endemic region for the presence and spread of NDM-type MBLs, and prevalence rates of Enterobacterales producing $bla_{NDM}$ were found in a range of >30% in hospitals of India and Pakistan (15, 16). In Pakistan, a study in Karachi reported that bacteria producing $bla_{NDM}$ were found to be responsible for 66% and 57% mortality in neonatal and adult patients, respectively (17). Similarly, in Pakistan, another study reported the death of four out of nine neonates due to bacteria producing $bla_{NDM}$ genes (18). Several other studies have been carried out to examine the dissemination of $bla_{NDM}$ (19–22) in Pakistan and reported an increase in the prevalence of these genes. In 2020, a meta-analysis reported a 28% pooled proportion of clinical carbapenem-resistant Gram-negative bacteria from Pakistan (23).

In this study, we performed whole genome sequencing of clinical *E. coli* isolates producing $bla_{NDM}$ genes and reported the detailed genetic context of $bla_{NDM}$-carrying plasmids. This knowledge provides insight into genetic characteristics and potential transmissions of the plasmids among clinical *E. coli* isolates in Pakistan.

## MATERIALS AND METHODS

### Isolation of carbapenem-resistant *E. coli* isolates

A total of 240 *E. coli* strains recovered from urine or pus cultures were collected from laboratories of two tertiary care hospitals in Faisalabad in 2019 and 2020 (24). These isolates were sub-cultured on CHROMagar media plates supplemented with 1 µg/mL meropenem and incubated overnight at 37℃ for purity checks and isolation of carbapenem-resistant *E. coli* isolates.

### Identification of bacteria

Matrix-assisted laser desorption ionization-TOF (MALDI-TOF) (Bruker Daltonics, Germany) was carried out for protein-based confirmation of bacteria at the species level.

## Molecular identification of the $bla_{NDM}$ gene

DNA of *E. coli* isolates was subjected to PCR to screen the presence of $bla_{NDM}$ gene using primers and conditions as described previously (25).

## Antimicrobial susceptibility testing

Antimicrobial susceptibility testing of all carbapenem-resistant *E. coli* isolates was performed by disk diffusion method using the Muller Hinton agar plates against 12 antimicrobials. Broth microdilution and E-tests were performed to analyze the susceptibility of isolates against colistin and tigecycline, respectively. *E. coli* strain ATCC 25922 was used as the control strain. Breakpoints of all antibiotics were interpreted according to EUCAST criteria (26).

## Pulsed-field gel electrophoresis analysis

All carbapenem-resistant *E. coli* isolates were further subjected to XbaI and S1 PFGE analysis as described by CDC 2016 (https://www.cdc.gov/pulsenet/pdf/ecoli-shigella-salmonella-pfge-protocol-508c.pdf). Briefly, agarose blocks of bacterial DNA were prepared in 1% SeaKem Gold Agarose (Lonza, Rockland, ME, USA) with 0.5X TBE (Tris–borate–ethylene) buffer, digested with XbaI and S1 nuclease enzymes and separated by electrophoresis using a CHEF Mapper XA Apparatus (Bio-Rad, Hercules, CA, USA) at 6 V $cm^{-1}$ at 14ºC, with an initial pulse time of 4 s and a final pulse time of 45 s for 22 h. In-gel DNA–DNA hybridization with a $bla_{NDM}$ probe labeled with $^{32}$P was carried out to determine the genetic location of carbapenem-resistant genes as described previously (27).

## Conjugation experiment

Conjugation experiments were carried out to determine the transferability of plasmid-mediated $bla_{NDM}$ genes using sodium azide-resistant *E. coli* J53 as the recipient strain, as previously described (28). Briefly, isolates were grown on chromogenic media plates with 1 µg/mL meropenem (AstraZeneca, London, UK) and *E. coli* J53 on chromogenic media with 200 µg/mL sodium azide. Pure cultures were propagated by picking a single colony and inoculating in 10 mL of LB broth for incubation at 37°C for 18 h, with shaking at 200 rpm. Mating was undertaken by mixing 1.5 mL of overnight culture of the resistant strain with 1 mL of *E. coli* J53 bacterial culture and 2 mL of LB broth. After incubation for 18 h at 37°C, 10 µL was used to inoculate plates containing 200 µg/mL sodium azide and 1 µg/mL meropenem to select transconjugants. Single colonies were subcultured and analyzed for the presence of resistance genes by PCR. Transfer frequencies were calculated by the colony forming unit (CFU) count of transconjugants against the CFU count of the donor.

## Whole-genome sequencing and bioinformatics analysis

Total gDNA was extracted from an overnight culture (2 mL) on a QIAcube automated system (Qiagen). Following extraction, gDNA was quantified by fluorometric methods using a Qubit (ThermoFisher Scientific, USA), with quality ratios of gDNA (A260/280 and 260/230) determined via Nanodrop (ThermoFisher Scientific, USA). Genomic DNA libraries are prepared for whole-genome sequencing using the NexteraXT Kit (Illumina), as described by the manufacturer. Paired-end sequencing was performed using the Illumina MiSeq platform (MiSeq Reagent V3 Kit; 2 × 300 cycles). For each *E. coli* isolate, at least 20 × coverage was generated. Raw sequence reads were trimmed using Trim Galore and the genomes were de novo-assembled into contigs using SPAdes (3.9.0) with a pre-defined kmers set (29). Raw reads were also assembled with Geneious (10.0.9; Biomatters Ltd.) *de novo* assembler, set at medium sensitivity for analysis of paired Illumina reads. Geneious was used to map both sets of contigs to reference genes identified by closest BLAST homology and to annotate genes from closest homologs in

NCBI Genome database. Resistance genes were identified using Resfinder within CGE, and plasmids were identified within the genome assembly and typed using Plasmidfinder (30).

To elucidate the genetic environments of $bla_{NDM}$ genes, 10 representative $bla_{NDM}$ isolates carrying plasmids were selected based on the phylogenetic analysis to perform MinION (Oxford Nanopore Technologies Ltd., Oxford Science Park, UK) sequencing to obtain the complete plasmid sequences. Large-scale bacterial gDNA was extracted. Two loops of a full bacterial colony in 9.5 mL of TE buffer were mixed with 50 µL proteinase K (20 mg/mL) and 0.5 mL of 10% SDS. After incubation at 37°C for 1 h, 1.8 mL heated 5M NaCl was added and was incubated at 65°C for 5 min. The suspension was treated with 1.5 mL of heated CTAB/NaCl and was incubated at 65°C for 20 min. Then an equal volume of chloroform was added to the mixture followed by gently shaking for 1 h. The mixture was centrifuged at 13,000 rpm for 15 min. The supernatant was taken, and an equal volume of isopropanol was added until the clumping of gDNA was visualized. DNA was washed with 70% ethanol, and 200 µL of water was added to dissolve the gDNA. DNA library was prepared by pooling all barcoded samples to aim for a final DNA concentration >500 ng/µL, and 1 µL of RAD was added to DNA. A final mixture of 75 µL (34 µL sequencing buffer, 30 µL water, and 11 µL DNA library) was loaded into the flow cell. MinION device was connected to MinKNOW GUI to obtain the reads. The raw data in fast5 format were base called with the high-accuracy mode and demultiplexed using Guppy 4.2.2 (31). Unicycler (0.4.4) was used to yield hybrid assembly using both Illumina short reads and minION long reads (32). This process included assembling the long reads with Flye v2.8 and following a 5-round polishing using Pilon (33) with the Illumina short reads of the same sample (34). Comparisons of the complete *E. coli* plasmids were visualized using BRIG and EasyFig (35).

### G. mellonella pathogenicity model

Pathogenicity of carbapenem-resistant *E. coli* isolates belonging to ST-405, ST361, and ST-167 was examined in a Galleria model. Larvae of the wax moth *G. mellonella* were used as an animal model for disease resulting from infection challenges of the test strains as described previously. Briefly, strains were standardized in suspensions equating to $1 \times 10^7$, $10^6$, and $10^5$ CFU/mL. Using a Hamilton Syringe, 10 µL of each suspension was injected into the hemocoel of the *G. mellonella* larvae, larvae were incubated in the dark at 37°C for 72 h, and the amount of dead and alive worms was checked every 24 h. Death was denoted when larvae no longer responded to touch. In addition to this, three more groups of 10 larvae were injected with non-pathogenic *E. coli* ATCC25922 to evaluate whether larvae were killed by non-infection related reactions, with 10 µL of PBS to measure any lethal effects due to physical injury and positive control inoculation with pathogenic strain KP-2 belonging to ST-131. Results were analyzed by Kaplan-Meier survival curves (GraphPad Prism statistics software) (36).

## RESULTS

### General characteristics, molecular identification, and antibiotic susceptibility of CR-EC isolates

Fifty-two non-repetitive carbapenem-resistant *E. coli* isolates were isolated from two hospitals over 1 year. These non-duplicated isolates were mainly isolated from urine (79%) followed by pus (21%) cultures. Similarly, the percentage of CR-EC isolated from female and male samples was found to be 73% and 27%, respectively. The age of patients ranged from 14 to 68 years, with a mean of 49 year. The highest number of isolates was found to be in the 40–49 age range (32%), while the lowest number of isolates was found in the 20–30 age range (6%).

Among the 52 carbapenem-resistant *E. coli* isolates, 34 strains were demonstrated to be $bla_{NDM}$ producers via specific $bla_{NDM}$ PCR. Antimicrobial susceptibility testing revealed that all 34 *E. coli* isolates were MDR strains, and they were resistant to multiple

categories of antibiotics ($n > 3$) (Table S1 in the supplemental material). Therefore, each isolate carried at least three categories of resistance genes associated with resistance phenotype (Table 1). Almost all these isolates were non-susceptible to fluoroquinolones (95–100%), cephalosporins (87–95%), and penicillins (82–97%). Moreover, over half of the isolates were resistant to aminoglycosides (57%). The MICs results of CR-EC isolates revealed 100% susceptibility to tigecycline and colistin.

## Pulsed-field gel electrophoresis

XbaI PFGE analysis of carbapenem-resistant isolates ($n = 34$) categorized them into five different clusters. Cluster A (PK-5027, PK-5034, PK-5068, and PK-5172) includes four isolates. Cluster B (PK-5092, PK-5171, and PK-5202) includes three isolates. Clusters C (PK-5093 and PK-5095), D (PK-5140 and PK-5141), and E (PK-5112 and PK-5198) include two isolates each. All other 21 isolates appeared to be singletons. S1-PFGE analysis showed that $bla_{NDM}$ was located on a plasmid with varying lengths. The number and sizes of plasmids in *E. coli* isolates were found to be between 1 and 5 and 40 kbps and 250kbps, respectively (Table S2).

## Conjugation

The conjugation experiment confirmed that $bla_{NDM}$ was able to transfer successfully into *E. coli* J53 strain. The range of conjugation frequencies for all isolates was observed between $1.59 \times 10^{-1}$ and $6.46 \times 10^{-8}$ per donor (Table S2).

## Resistance determinants

Analysis of antibiotic resistance genes revealed the presence of $bla_{NDM-5}$ in 91% ($n = 31/34$), while the percentage prevalence of $bla_{NDM-1}$, $bla_{NDM-20}$, and $bla_{NDM-21}$ was 3% ($n = 1/34$) each. Between 4 and 20 antibiotic resistance genes were found in each isolate. PK-5171 was found to harbor 20 antibiotic-resistance genes belonging to nine different classes of antibiotics, followed by two isolates PK-5127 and PK-5136 and four isolates PK-5081, PK-5112, PK-5138, and PK-5198 found to harbor 17 and 16 genes, respectively. The 34 sequenced isolates harbored a plethora of resistant genes, including β-lactamases $bla_{OXA-1}$ , $bla_{CTX-M-15}$, $bla_{EC-15}$, $bla_{TEM-1}$, $bla_{EC-8}$, $bla_{EC}$, $bla_{CMY-145}$, $bla_{EC-18}$, $bla_{CTX-M-139}$, $bla_{CTX-M-101}$, $bla_{OXA-181}$, $bla_{CMY-42}$, $bla_{CMY-102}$, $bla_{CMY-131}$, $bla_{OXA-10}$, and $bla_{CTX-M-103}$, aminoglycosides (*aadA2* 65%, *aac(6')-Ib-cr*, *aadA5*, *aadA1*, *aph(3")-Ib*, and *aph(6)-Id*, *aac(3)-IId*, *aadA11*, *aadA16*, and *aac(3)-IIa*, fluoroquinolones (*qnrS1*, *qnrB6*, *qepA1*, *qepA6*, *qepA8*, and *qepA9*), sulfonamides (*sul1*, *sul2*, and *sul3*), trimethoprim (*dfrA12*, *dfrA14*, and *dfrA17*), phenicols (*catB3*, *catA1*, and *cmlA1*), tetracyclines (*tet(A)*, *tet(B)*, and *tet(34))*, and macrolides (*mph(A)*). Genes $bla_{OXA-1}$ (73%, $n = 25/43$) and $bla_{CTX-M-15}$ (50%, $n = 17/34$) were found to be the most predominant β-lactamases among the isolates. All isolates harbored more than one *bla* in different combinations. Aminoglycoside resistance was mostly mediated by *aadA2* (65%, $n = 22/34$) and *aac(6')-Ib-cr* (47%, $n = 16/34$) that confers resistance to both aminoglycosides and fluoroquinolones. Among the tetracycline resistant genes, Gene *tet(A)* (67%, $n = 23/34$) was the most common tetracycline resistance gene carried by isolates (Table 1).

## Virulence genes

A total of 17 various virulence genes were detected across 34 *E. coli* isolates in different combinations of 4 to 16 genes. The most frequent genes found among the isolates were *terC* (100%, $n = 34/34$), *traT* (88%, $n = 30/34$), *gad* (79%, $n = 27/34$), *fyuA* and *irp2* (61%, $n = 21/34$), *capU* (52%, $n = 18/34$), and *sitA* (50%, $n = 17/34$). Isolates detected with more than 10 virulence genes PK-5138 ($n = 16$), PK-5136 ($n = 15$), PK-5081 and PK-5127 ($n = 14$), PK-5178 ($n = 13$), and PK-5179 ($n = 12$) belong to ST-405. Only two isolates PK-5171 and PK-5037, with virulence genes of 11 and 10, respectively, belong to ST-167 (Table 1).

**TABLE 1** Genomic characteristics of clinical *E. coli* isolates harboring *bla*$_{NDM}$ gene

| Isolate ID | MLST | Inc types of plasmids | Resistance genes | Virulence genes |
|---|---|---|---|---|
| PK-5027 | ST-361 | IncFII, IncY, IncFIA, IncI1 | *tet(A), sul1, bla*$_{EC}$*, bla*$_{OXA-1}$*, bla*$_{CMY-145}$*, aadA2, aadA1, bla*$_{NDM-5}$*, mph(A), dfrA12, qepA1, qepA8* | *capU, gad, sitA, terC, traT* |
| PK-5034 | ST-361 | IncFIA, IncFII, IncY, IncI1 | *tet(A), sul1, bla*$_{EC}$*, bla*$_{CMY-102}$*, bla*$_{OXA-1}$*, aadA2, aadA1, bla*$_{NDM-5}$*, mph(A), dfrA12, qepA8, qepA6* | *capU, gad, sitA, terC, traT* |
| PK-5037 | ST-1702 | IncFIA, IncI1, ColRNAI, IncFIC | *tet(34), sul1, bla*$_{EC-15}$*, bla*$_{CMY-42}$*, aadA2, bla*$_{NDM-5}$*, dfrA12* | *capU, gad, cia, cib, iss, fyuA, hra, iss, terC, traT, irp2* |
| PK-5052 | ST-405 | IncFIB, IncFIA, IncFII, p0111 | *tet(34), tet(A), sul1, bla*$_{EC-8}$*, bla*$_{CTX-M-15}$*, bla*$_{OXA-1}$*, catB3, aadA2, aac(6')-Ib-cr, bla*$_{NDM-5}$*, mph(A), dfrA12* | *chuA, kpsE, kpsMII_K5, fyuA, terC, traT, irp2* |
| PK-5055 | ST-1702 | ColRNAI, IncFIA, IncFIC | *tet(34), sul1, bla*$_{EC-15}$*, bla*$_{TEM-1}$*, aadA5, bla*$_{NDM-5}$*, dfrA12, dfrA17* | *capU, irp2, fyuA, gad, iss, hra, terC, traT* |
| PK-5068 | ST-361 | IncFII | *tet(A), sul1, bla*$_{EC}$*, bla*$_{CMY-145}$*, catA1, catA1, aadA2, aadA1, bla*$_{NDM-5}$*, mph(A), dfrA12, qepA8* | *capU, gad, sitA, terC, traT* |
| PK-5081 | ST-405 | IncFIA, IncFIB, p0111, IncFIC, Col(MG828) | *tet(34), tet(A), sul1, sul2, bla*$_{EC-8}$*, bla*$_{CTX-M-15}$*, bla*$_{TEM-1}$*, bla*$_{OXA-1}$*, catB3, aadA5, aph(3")-Ib, aph(6)-Id, aac(6')-Ib-cr, bla*$_{NDM-5}$*, mph(A), dfrA17* | *kpsE, kpsMII_K5, chuA, eilA, fyuA, sat, sitA, iucC, iutA, papA_F43, terC, traT, iha, irp2* |
| PK-5092 | ST-167 | IncFIC, IncFIA, IncFIB, Col156, Col(BS512) | *tet(34), tet(A), sul1, bla*$_{EC-15}$*, bla*$_{CTX-M-15}$*, bla*$_{OXA-1}$*, catB3, aadA2, aac(6')-Ib-cr, bla*$_{NDM-5}$*, mph(A), dfrA12* | *capU, celb, gad, hra, iss, terC, traT* |
| PK-5093 | ST-156 | IncFIB, Col(MGD2), ColRNAI, IncFIC | *tet(B), tet(34), sul1, bla*$_{EC-18}$*, bla*$_{TEM-1}$*, aadA2, bla*$_{NDM-5}$*, dfrA12, qepA9* | *iss, lpfA, fyuA, gad, irp2, traT, papC, terC* |
| PK-5095 | ST-156 | IncFIB, Col(MGD2), ColRNAI, IncFIC | *tet(34), tet(B), sul1, bla*$_{EC-18}$*, bla*$_{TEM-1}$*, aadA2, bla*$_{NDM-5}$*, dfrA12, qepA9* | *iss, lpfA, papC, fyuA, gad, irp2, traT, terC* |
| PK-5096 | Unknown | IncFIA, IncFII, ColRNAI, IncI1 | *tet(34), tet(A), sul1, sul2, bla*$_{EC-15}$*, bla*$_{CMY-131}$*, aadA2, aph(3")-Ib, aph(6)-Id, bla*$_{NDM-5}$*, mph(A), dfrA12* | *capU, iss, terC, traT, hra, irp2, fyuA, gad* |
| PK-5099 | ST-361 | IncFII_1, IncFIA_1, IncI1 | *sul1, bla*$_{EC}$*, bla*$_{OXA-1}$*, catA1, aadA1, aadA2, bla*$_{NDM-20}$*, dfrA12, qepA1, qepA8* | *capU, gad, sitA, terC, traT* |
| PK-5112 | ST-167 | IncFIC, IncFIA, ColKP3, ColRNAI, Col(BS512) | *tet(34), tet(A), sul1, bla*$_{EC-15}$*, bla*$_{CTX-M-15}$*, bla*$_{OXA-181}$*, bla*$_{TEM-1}$*, bla*$_{OXA-1}$*, catB3, aadA2, aac(6')-Ib-cr, bla*$_{NDM-5}$*, mph(A), dfrA12, ere(A), qnrS1* | *capU, fyuA, gad, irp2, iss, terC, traT* |
| PK-5116 | ST-167 | IncFIA, IncFII | *tet(A), sul1, bla*$_{OXA-1}$*, bla*$_{CTX-M-15}$*, catB3, aac(6')-Ib-cr, aadA5, bla*$_{NDM-5}$*, mph(A), dfrA17* | *capU, gad, hra, terC, traT, iss, hlyE* |
| PK-5127 | ST-405 | IncFIA, IncFIC, Col(MG828) | *tet(34), tet(A), sul1, sul2, bla*$_{EC-8}$*, bla*$_{CTX-M-139}$*, bla*$_{CTX-M-101}$*, bla*$_{TEM-1}$*, bla*$_{OXA-1}$*, catB3, aadA5, aph(6)-Id, aph(3")-Ib, aac(6')-Ib-cr, bla*$_{NDM-5}$*, mph(A), dfrA17* | *afaD, chuA, fyuA, iha, irp2, iucC, iutA, kpsE, kpsMII_K5, papA_F43, sat, sitA, terC, traT* |
| PK-5136 | ST-405 | IncFIC, Col(MG828), IncFIA | *tet(34), tet(A), sul1, sul2, bla*$_{EC-8}$*, bla*$_{CTX-M-101}$*, bla*$_{TEM-1}$*, bla*$_{CTX-M-103}$*, bla*$_{OXA-1}$*, catB3, aadA5, aph(6)-Id, aph(3")-Ib, aac(6')-Ib-cr, bla*$_{NDM-5}$*, mph(A), dfrA17* | *iha, irp2, iucC, afaD, chuA, papA_F43, sat, sitA, fyuA, gad, iutA, terC, traT, kpsE, kpsMII_K5* |
| PK-5138 | ST-405 | IncFIC, IncFIA, p0111, Col(MG828) | *tet(34), tet(A), sul1, sul2, bla*$_{EC-8}$*, bla*$_{CTX-M-15}$*, bla*$_{OXA-1}$*, bla*$_{TEM-1}$*, catB3, aadA5, aph(3")-Ib, aph(6)-Id, aac(6')-Ib-cr, bla*$_{NDM-5}$*, mph(A), dfrA17* | *fyuA, gad, iha, irp2, afaD, chuA, eilA, iucC, papA_F43, sat, sitA, terC iutA, kpsE, kpsMII_K5, traT* |
| PK-5140 | ST-405 | IncFIA, IncFIB, IncFII | *tet(34), tet(B), sul1, bla*$_{EC-8}$*, bla*$_{TEM-1}$*, bla*$_{CTX-M-15}$*, bla*$_{OXA-1}$*, catB3, aadA2, aadA5, aac(6')-Ib-cr, aac(3)-IId, bla*$_{NDM-5}$*, dfrA12, dfrA17* | *chuA, irp2, sitA, terC, fyuA, kpsE, kpsMII_K5, gad, hra* |
| PK-5141 | ST-405 | IncFIA, IncFIB, IncFII | *tet(34), tet(B), sul1, bla*$_{EC-8}$*, bla*$_{CTX-M-15}$*, bla*$_{OXA-1}$*, catB3, aadA2, aac(6')-Ib-cr, aac(3)-IId, aadA5, bla*$_{NDM-5}$*, dfrA12, dfrA17* | *chuA, fyuA, hra, irp2, kpsMII_K5, sitA, terC* |

(*Continued on next page*)

**TABLE 1** Genomic characteristics of clinical *E. coli* isolates harboring *bla*$_{NDM}$ gene (*Continued*)

| Isolate ID | MLST | Inc types of plasmids | Resistance genes | Virulence genes |
|---|---|---|---|---|
| PK-5144 | ST-405 | IncFIC, IncFIA, IncFIB | *tet(34), tet(A), sul1, bla*$_{EC-8}$*, bla*$_{CTX-M-15}$*, bla*$_{OXA-1}$*, catB3, aadA2, aac(6')-Ib-cr, bla*$_{NDM-5}$*, mph(A), dfrA12* | *afaD, chuA, fyuA, irp2, kpsE, kpsMII_K5, terC, traT* |
| PK-5151 | ST-2450 | IncX3, ColRNAI, IncFIB, p0111 | *tet(34), sul2, bla*$_{EC-15}$*, aph(6)-Id, aph(3")-Ib, aac(3)-IIa, bla*$_{NDM-5}$ | *capU, fyuA, gad, irp2, iss, sitA, terC* |
| PK-5152 | ST-405 | IncFIA, IncFIB, IncFII | *tet(A), sul1, bla*$_{CTX-M-15}$*, bla*$_{OXA-1}$*, catB3, aac(6')-Ib-cr, bla*$_{NDM-5}$*, mph(A), dfrA12* | *kpsMII_K5, chuA, fyuA, gad, kpsE, terC, traT, irp2* |
| PK-5160 | Unknown | Col156, IncFIB, IncFII, IncI1 | *sul1, bla*$_{OXA-10}$*, cmlA1, bla*$_{NDM-1}$ | *celb, gad, lpfA, terC* |
| PK-5171 | ST-167 | IncFIA, IncR, IncFIB, IncFIC, ColRNAI | *tet(34), tet(B), tet(A), sul1, bla*$_{EC-15}$*, bla*$_{CTX-M-139}$*, bla*$_{OXA-1}$*, catA1, catB3, aadA16, aac(6')-Ib-cr, aadA2, aadA5, bla*$_{NDM-5}$*, mph(A), dfrA27, dfrA12, dfrA17, qnrS1, qnrB6* | *capU, fyuA, gad, hra, irp2, iss, iucC, iutA, sitA, terC, traT* |
| PK-5172 | ST-361 | IncI1, IncFIA, IncFII | *tet(A), sul1, bla*$_{EC}$*, bla*$_{CMY-102}$*, bla*$_{OXA-1}$*, bla*$_{CMY-145}$*, catA1, aadA1, aadA2, bla*$_{NDM-5}$*, mph(A), dfrA12, qepA8* | *capU, gad, sitA, terC, traT* |
| PK-5176 | ST-167 | IncFIC, IncFIA, IncX3, Col(MG828) | *tet(34), tet(A), sul1, bla*$_{OXA-1}$*, bla*$_{EC-15}$*, bla*$_{CTX-M-15}$*, catB3, aac(6')-Ib-cr, aadA5, bla*$_{NDM-5}$*, mph(A), dfrA17* | *capU, gad, hra, iss, terC, traT* |
| PK-5178 | ST-405 | IncFIC, IncFIA, p0111, IncFIB, Col(MG828) | *tet(34), tet(A), sul1, sul2, bla*$_{EC-8}$*, bla*$_{CTX-M-15}$*, bla*$_{OXA-1}$*, bla*$_{TEM-1}$*, catB3, aadA5, aph(6)-Id, aph(3")-Ib, aac(6')-Ib-cr, blaNDM-5, mph(A), dfrA17* | *chuA, fyuA, iha, irp2, iucC, iutA, kpsE, kpsMII_K5, papA_F43, sat, sitA, terC, traT* |
| PK-5179 | ST-405 | Col(BS512), IncFII | *tet(B), sul1, bla*$_{CTX-M-15}$*, aadA2, bla*$_{NDM-5}$*, dfrA12* | *chuA, fyuA, gad, eilA, irp2, kpsE, kpsMIII_K98, papA_F43, papC, sitA, terC, traT* |
| PK-5196 | ST-361 | IncFII, IncY, IncFIA, IncI1 | *tet(A), sul1, bla*$_{EC}$*, bla*$_{CMY-145}$*, bla*$_{OXA-1}$*, catA1, aadA11, aadA2, aadA1, bla*$_{NDM-5}$*, mph(A), dfrA12, qepA8* | *capU, gad, sitA, terC, traT* |
| PK-5198 | ST-167 | IncFIA, IncFIC, ColKP3, ColRNAI, Col(BS512) | *tet(34), tet(A), sul1, bla*$_{EC-15}$*, bla*$_{CTX-M-15}$*, bla*$_{OXA-181}$*, bla*$_{TEM-1}$*, bla*$_{OXA-1}$*, catB3, aadA2, aac(6')-Ib-cr, bla*$_{NDM-5}$*, mph(A), dfrA12, ere(A), qnrS1* | *capU, fyuA, gad, irp2, iss, terC, traT* |
| PK-5202 | ST-167 | ColRNAI, IncFIA, IncFIC | *tet(34), sul1, sul2, bla*$_{EC-15}$*, bla*$_{CTX-M-15}$*, aadA2, aph(6)-Id, aph(3")-Ib, bla*$_{NDM-5}$*, dfrA12* | *capU, fyuA, gad, hra, irp2, terC, traT* |
| PK-5209 | ST-361 | IncI1, IncFII | *tet(A), bla*$_{EC}$*, bla*$_{OXA-1}$*, catA1, aadA1, bla*$_{NDM-21}$*, qepA1, qepA6* | *capU, gad, sitA, terC, traT* |
| PK-5224 | ST-2851 | Col156, IncFIA, ColRNAI, IncFII | *tet(A), sul1, bla*$_{EC-15}$*, bla*$_{CTX-M-15}$*, bla*$_{TEM-1}$*, aadA2, bla*$_{NDM-5}$*, dfrA12* | *celb, gad, hra, lpfA, terC, traT* |
| PK-5238 | ST-2851 | IncI1, IncFIA, IncFII, Col(MG828) | *tet(A), sul1, bla*$_{EC-15}$*, bla*$_{CTX-M-15}$*, bla*$_{CMY-42}$*, aadA2, bla*$_{NDM-5}$*, mph(A), dfrA12* | *gad, hra, lpfA, terC, traT* |

## Replicon typing

Screening of plasmid replicons among 34 *E. coli* isolates using the PlasmidFinder database detected nine plasmid replicons, including FII, FIA, FIB, FIC, X3, R, Y, Col, and p0111. FIA was the predominant replicon identified in 80% ($n = 27/34$) of isolates followed by Col, FII, FIC, and FIB replicon types, with 61% ($n = 21/34$), 52% ($n = 18/34$), 47% ($n = 16/34$), and 38% ($n = 13/34$), respectively. There were 1 ($n = 1$), 2 ($n = 3$), 3 ($n = 10$), 4 ($n = 14$), and 5 ($n = 6$) plasmids detected in the isolates (Table 1).

## Phylogenetic analysis

WGS analysis provided comprehensive information for the 34 *bla*$_{NDM}$ carrying *E. coli* and their phylogenetic relationship. Phylogenetic relationships among *E. coli* isolates were determined by using the online tool CSI Phylogeny (1.4 version) (https://cge.cbs.dtu.dk/

services/CSIPhylogeny/). The pipeline comprises of various freely available programs. The paired-end reads from each isolate were aligned against the reference genome, using Burrows-Wheeler Aligner (BWA), SAMtools, "mpileup" command, and bedtools. The single nucleotide polymorphism (SNP) based phylogenetic tree was generated by calling and filtering SNPs, site validation, and phylogeny based on a concatenated alignment of the high-quality SNPs. For inferring phylogeny, the analysis was run with the standard parameters, and the NCTC11129 strain genome (GenBank accession number NZ_LR134222.1) was used as a reference sequence. The analysis was run with the default parameters: a minimal depth at SNP positions was 10 reads along with a relative depth at SNP positions of 10%, a minimal distance between SNPs was of 10 bp, a minimal SNPs quality was 30, a minimal Z-score was of 1.96, a minimal SNP quality was 30, and a minimal read mapping quality was of 25. The Z-score expresses the confidence with which a base was called at a given position, and the Z-score was 1.96. Number of SNPs exhibited among closely related isolates was calculated using a distance matrix file generated as a result of phylogeny (37). The output core alignment file was used to construct the Maximum-likelihood tree with 1,000 bootstrap replications, by using MEGA-X v 10.0.5 (https://www.megasoftware.net/home) (38). The phylogenetic tree of the alignments was visualized and edited by iTOL v 4.4.2 software (https://itol.embl.de) (39). Genomes of the sequenced isolates covered 70% of the NCTC11129 reference genome. The phylogenetic tree showed that among 34 distinct *E. coli* strains, 2 belong to unknown STs and 32 belong to 8 ST types: ST405 ($n = 11$), ST167 ($n = 7$), ST361 ($n = 7$), ST156 ($n = 2$), ST2851 ($n = 2$), ST1702 ($n = 2$), and ST2450 ($n = 1$) (Fig. 1). Isolates exhibiting the same sequence types are grouped closely in the resulting tree. Notably, the reference genome was grouped with three isolates belong to ST156 and unknown ST. Interestingly, two isolates PK-5055 and PK-5037 from ST-1702 and one isolate of unknown ST grouped with isolates belong to ST167. We noticed that two *E. coli* strains belonging to unknown STs were isolated from urine culture. The other 25 and 7 *E. coli* strains isolated from urine and pus cultures, respectively, were distributed among all eight sequence types. We further determined the SNPs distance of the core genome. SNPs matrix in the final dataset revealed a minimum of six SNPs and a maximum of 41,296 SNPs detected between all examined genomes. The core alignment showed that in ST405, isolates PK5140 and PK5141 differed from each other by nine SNPs. In ST361, there was a difference of 26 SNPs between the isolates PK-5027 and PK-5034 and 5099 and 5172. Similarly, in ST167 isolates 5112 and PK-5198 and in ST2851, isolates PK5224 and PK5238 differed by 28 and 71 SNPs (Table S3).

## Comparative analysis of plasmids

Genomic DNA of 10 isolates (PK-5055, PK-5081, PK-5099, PK-5112, PK-5144, PK-5160, PK-5171, PK-5172, PK-5209, and PK-5224) were selected according to phylogenetic analysis to be sequenced with the MinION long-read platform for comparative analysis of plasmids. Basic information on $bla_{NDM}$ harboring plasmids in these isolates was summarized in Table 2. Isolates PK-5055, PK-5224, and PK-5209 harbored $bla_{NDM-5}$, $bla_{NDM-5}$, and $bla_{NDM-21}$ gene, respectively, on IncFII-IncFIA plasmid. In isolates PK-5081 and PK-5171, $bla_{NDM-5}$ gene was located on IncFII-IncFIA-IncFIB plasmid. Similarly, PK-5160, PK-5172, PK-5099, and PK-5144 harbored $bla_{NDM-1}$, $bla_{NDM-5}$, $bla_{NDM-20}$, and $bla_{NDM-5}$ gene, respectively, on IncHI2, IncFII, IncFIA, and IncFIB plasmids, respectively. In PK-5112, the $bla_{NDM-5}$ gene was found on the ColKP3-IncFII-IncFIA plasmid (Table 2). BLASTn analysis revealed that pPK-5099, pPK-5209, pPK-5224, pPK-5112, pPK-5055, and pPK-5172 shared homology with plasmid p52148-NDM-5 (Accession no. CP050384.1) of *E. coli* strain 52148 isolated from urine sample of human in 2019 in Prague with identity 99.70%, 99.23%, 99.63%, 99.91%, 99.90%, and 98.58% at coverage 100%, 96%, 91%, 80%, 74%, and 62%, respectively (Fig. 2A). BLAST search of pPK-5081 and pPK-5144 exhibited identity 99.94% and 99.89% at coverage of 98% and 87% with plasmid p_dm655_NDM5 (Accession no. CP095638.1) of *E. coli* strain dm655 isolated from human blood sample in Bangladesh in 2017 and plasmid pNDM_P30_L1_05.20 (Accession no. CP085061.1)

of *E. coli* strain P30_L1_05.20 isolated from human rectal sample in the UK in 2020, respectively (Fig. 2B). Similarly, BLASTn analysis of pPK-5171 revealed its 99.93% identity with p675SK2_B (Accession No. CP027703.1),and pMB5823_1 plasmid (Accession No. CP103646.1) at 98% and 85% of coverage, respectively (Fig. 2C). Plasmids p675SK2_B and pMB5823_1 of *E. coli* strains 675SK2 and 961 were isolated from wastewater in Switzerland in 2016 and blood samples of humans in the USA in 2018, respectively. IncHI2 type $bla_{NDM-1}$ bearing plasmid pPK-5160-$bla_{NDM-1}$ was detected in the PK-5160 strain. BLASTn analysis of pPK-5160 against the NCBI nr database showed that pPK-5160-$bla_{NDM-1}$ shared 99.71% identity at 86% and 85% coverage with plasmids pXJW9B277-HI2-N (Accession No. CP068042.1) and pL1 (Accession No. CP071712.1) of *E. coli* strains XJW9B277, and EC20017429 isolated from bovine cell culture in China in 2018 and gastroenteritis sample of human in Canada in 2017, respectively (Fig. 2D).

## Genetic environment of $bla_{NDM}$ gene

The genetic environment around the $bla_{NDM}$ gene located on IncF plasmids can be classified into three types of groups. These regions carrying by $bla_{NDM}$ gene were surrounded by IS*26*. For group

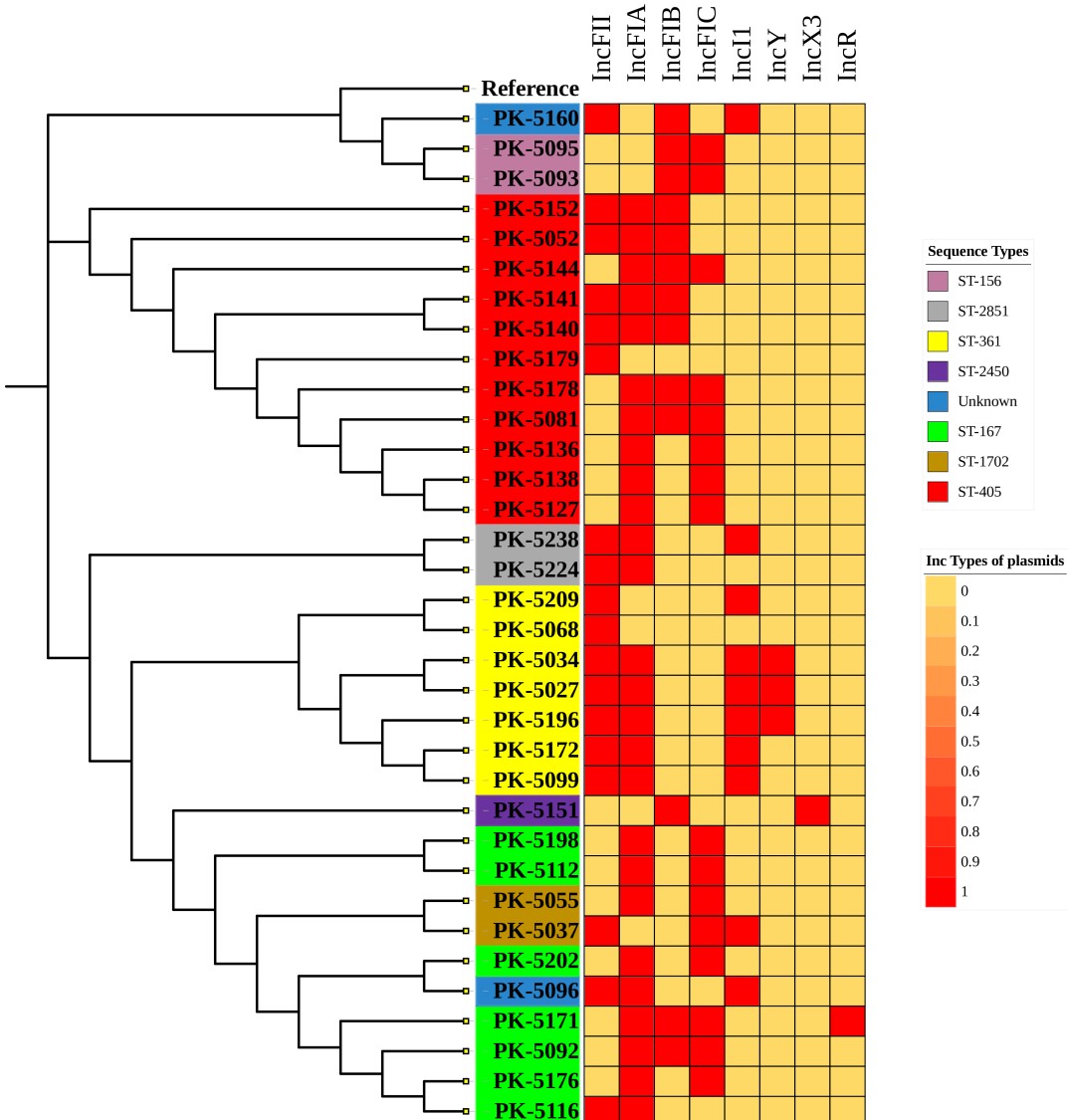

**FIG 1** Phylogenetic tree of all 34 $bla_{NDM}$-positive *E. coli* isolates.

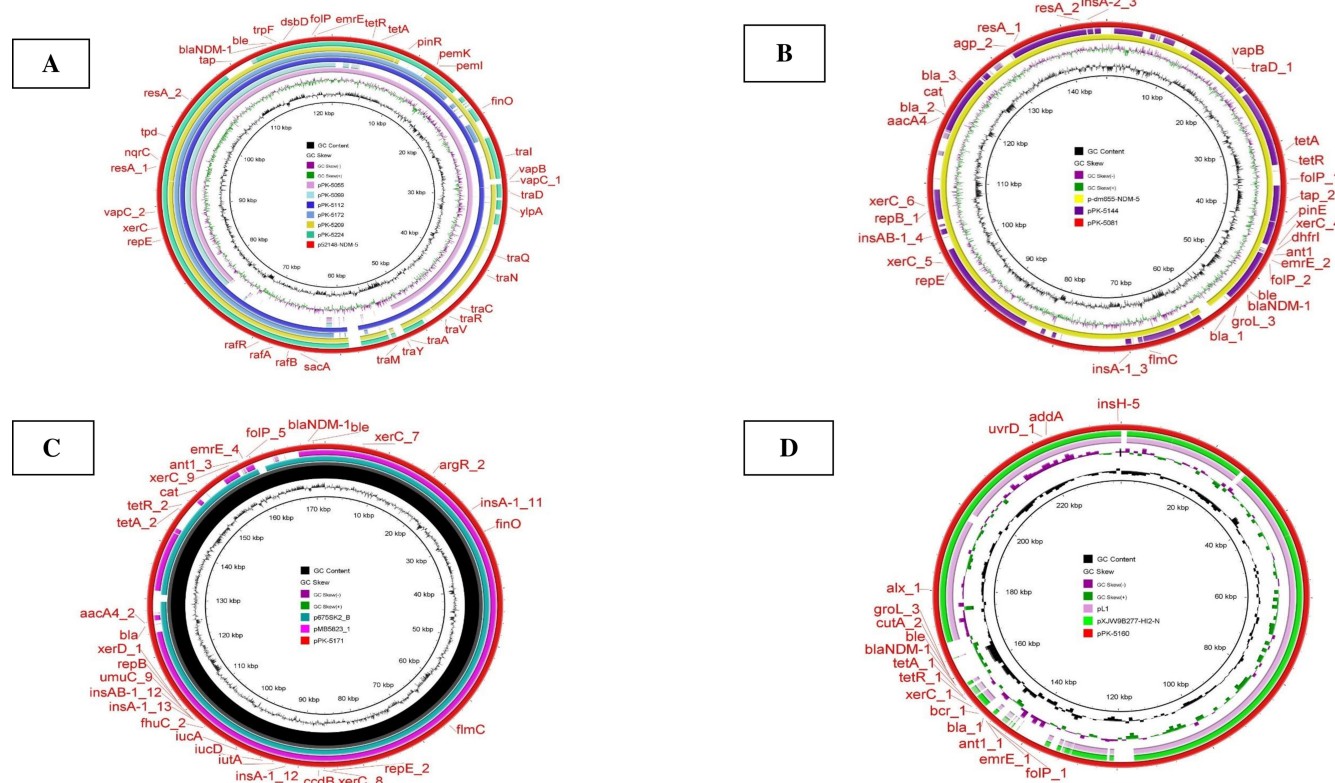

**FIG 2** (A, B, C, and D). Circular comparison of different IncF plasmids bearing *bla*<sub>NDM</sub> gene.

A (IS26-ΔISAba125-bla<sub>NDM</sub>-ble<sub>MBL</sub>-trpF-ISCR1-sul1-qacE-aadA2-dfrA12-IntI1-IS26-ΔTnAS1), the flanking genetic structure of *bla*<sub>NDM</sub> gene was composed of an *IS26* and incomplete *ISAba125* interrupted by insertion sequence ISCR1 and the genes *ble*<sub>MBL</sub> (bleomycin resistance), *trpF* (phosphoribosylanthranilate isomerase), *sul1* (sulfonamide-resistant dihydropteroate synthase), *qacE* (quaternary ammonium compound resistance protein), *aadA2* (aminoglycoside nucleotidyltransferase), *dfrA12* (dihydrofolate reductase), and *IntI1* (class one integron integrase) located downstream. Furthermore, this group also has downstream addition of *ΔTnAS1* associated with *IS26*. Compared with group A, group B (*IS26-IS26-ΔISAba125-bla*<sub>NDM</sub>-*ble*<sub>MBL</sub>-*trpF-ISCR1-sul1-qacE-aadA2-dfrA12-IntI1-IS26-ΔTnAS1*) includes another *IS26* upstream addition and completed downstream deletion of *ΔTnAS1*. Group C (*IS26-ΔISAba125-bla*<sub>NDM</sub>-*ble*<sub>MBL</sub>-*trpF-ISCR1-sul1-qacE-aadA2-dfrA12-IntI1-ΔTnAS3-IS26*) includes the downstream addition of *ΔTnAS3* associated with oppositely directed *IS26* (Fig. 3).

**TABLE 2** Characteristics of *E. coli* strains harboring *bla*<sub>NDM</sub> gene

| Isolate ID | Plasmid | Plasmid type | G + C content | Size (bp) |
|---|---|---|---|---|
| PK-5055 | pPK-5055-*bla*<sub>NDM-5</sub> | IncFII-IncFIA | 53.8% | 138,592 |
| PK-5081 | pPK-5081-*bla*<sub>NDM-5</sub> | IncFII-IncFIA-IncFIB | 51.5% | 145,982 |
| PK-5099 | pPK-5099-*bla*<sub>NDM-20</sub> | IncFIA | 54.7% | 45,613 |
| PK-5112 | pPK-5112-*bla*<sub>NDM-5</sub> | ColKP3-IncFII-IncFIA | 52.6% | 160,549 |
| PK-5144 | pPK-5144-*bla*<sub>NDM-5</sub> | IncFIB | 52.4% | 166,610 |
| PK-5160 | pPK-5160-*bla*<sub>NDM-1</sub> | IncHI2 | 45.9% | 238,344 |
| PK-5171 | pPK-5171-*bla*<sub>NDM-5</sub> | IncFII-IncFIA-IncFIB | 51.9% | 171,801 |
| PK-5172 | pPK-5172-*bla*<sub>NDM-5</sub> | IncFII | 51.6% | 126,392 |
| PK-5209 | pPK-5209-*bla*<sub>NDM-21</sub> | IncFII-IncFIA | 52.9% | 121,303 |
| PK-5224 | pPK-5224-*bla*<sub>NDM-5</sub> | IncFII-IncFIA | 52.7% | 102,257 |

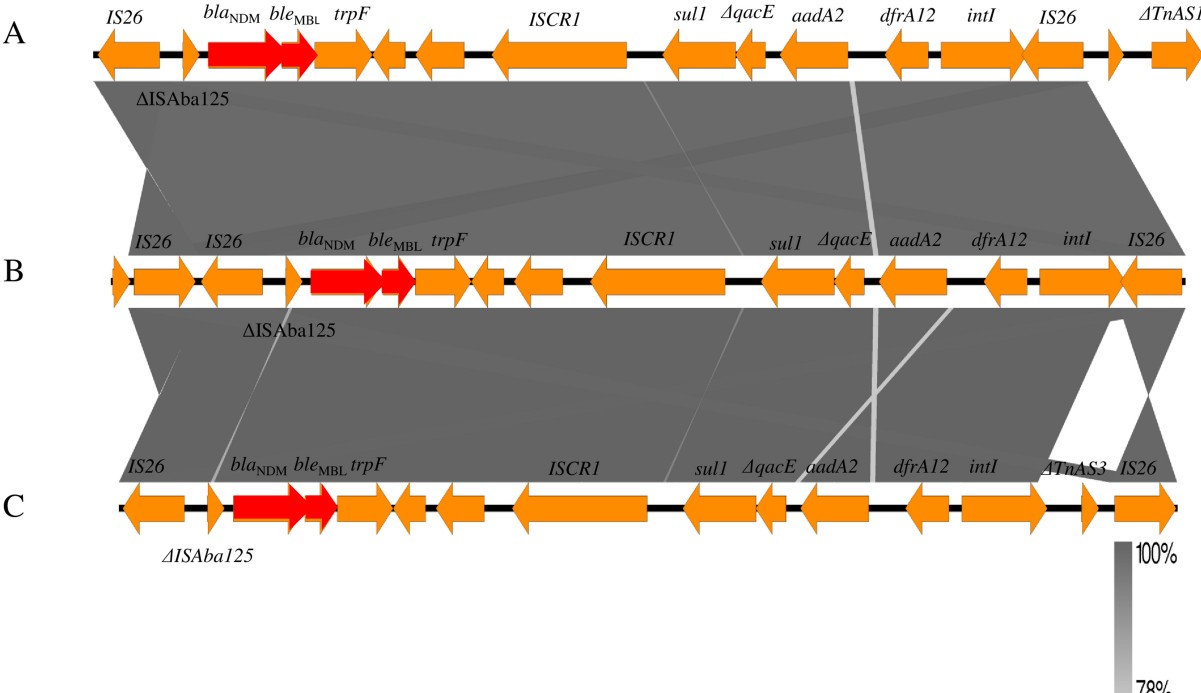

**FIG 3** Three (A–C) major types of $bla_{NDM}$-bearing genetic contexts among the $bla_{NDM}$-bearing IncF-type plasmids.

One IncHI2 type $bla_{NDM-1}$ bearing plasmid pPK-5160-$bla_{NDM-1}$ was also identified in the PK-5160 strain. The core genetic environment of the $bla_{NDM-1}$ gene in this plasmid contains multiple antibiotic resistance genes including *tetR*, *tetA*, $bla_{NDM-1}$, and $ble_{MBL}$ surrounded by IS26 and IS3000 upstream and downstream from $bla_{NDM-1}$ (Fig. 4).

## Clinical *E. coli* isolates and *G. mellonella* mortality

Pathogenicity of all clinical *E. coli* strains belonging to the ST-405 (*n* = 11), ST-361 (*n* = 8), and ST-167 (*n* = 7) was determined in *G. mellonella* larvae, dose titration was performed with culture $10^5$ to $10^7$ colony forming units. Two groups of 10 larvae were also injected with PBS and KP-2, a pathogenic strain, as a negative and positive control, respectively. Percentage survival of worms was observed for post-infection 72 h. Dose dependent survival was observed, as inoculum concentration $10^7$ killed more larvae than $10^6$ and $10^5$. PBS control injected larvae all remained alive over the 3 d time course. In contrast, KP-2 killed 80% of larvae during the time course. Larvae injected with isolates of ST-405, ST-167, and ST-351 showed more than 70%, 70%, and 80% survival, respectively (Fig. 5).

## DISCUSSION

AMR has been referred to as *"the silent tsunami facing modern medicine."* This study was designed to provide insights into the genomic epidemiology of clinical *E. coli* isolates in hospital settings in Pakistan by a whole-genome sequencing approach. *E. coli* is responsible for causing multiple community-acquired and nosocomial infections, including bacteremia, UTI, septicemia, wounds, and catheter-associated infections (40).

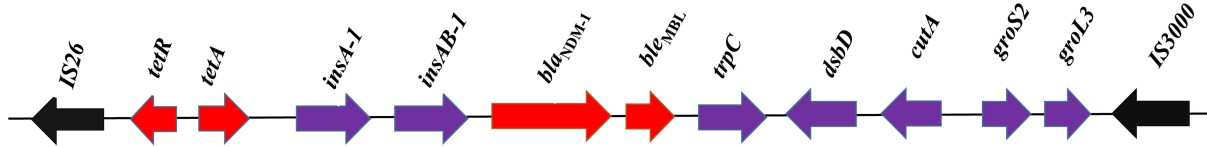

**FIG 4** Genetic elements surrounding the $bla_{NDM}$ gene on IncHI2 plasmid.

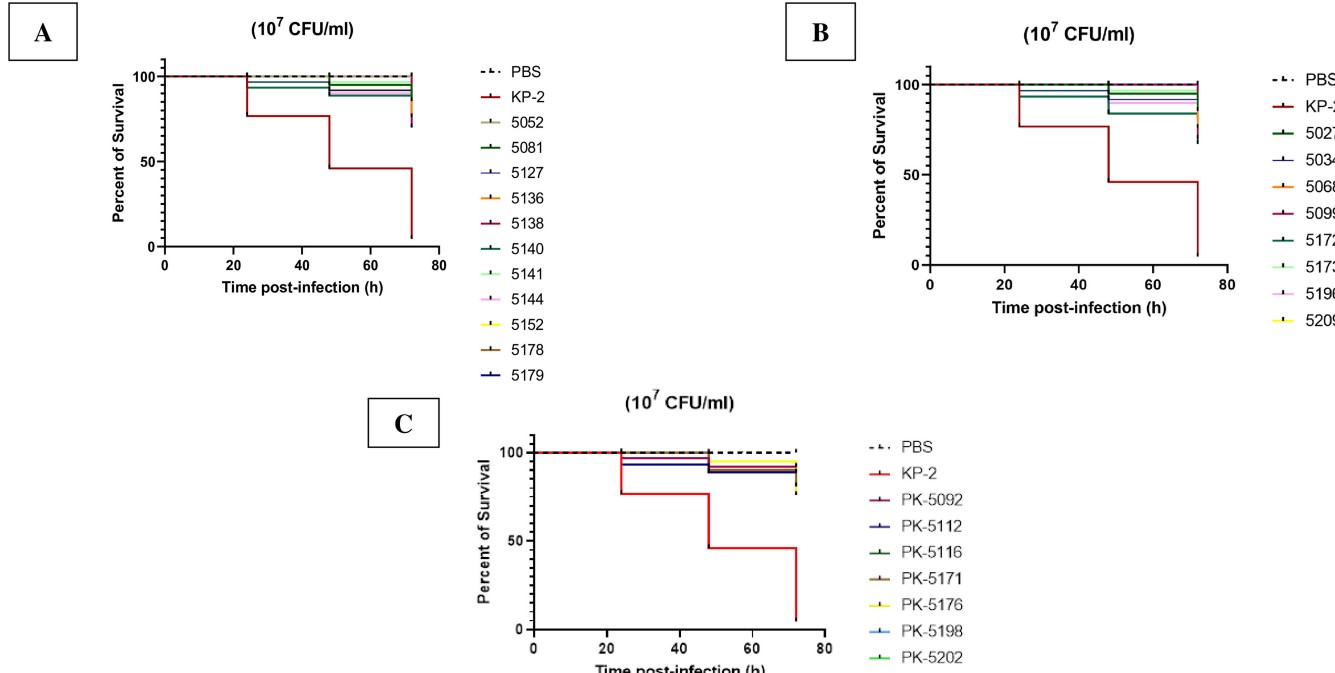

**FIG 5** Kaplan–Meier curves A, B, and C show the percentage of *G. mellonella* survival for 72 h post-infection with carbapenem-resistant strains of *E. coli* belonging to ST-405, ST-167, and ST-361, respectively.

In the current study, the majority of *E. coli* isolates were isolated from samples of urine (79%) and pus (21%). These findings agree with another study in Pakistan that found the prevalence of ExPEC isolates at 62.7% and 24.3% from urine and pus samples, respectively (24). Another study performed in Pakistan showed that *E. coli* causes 73% of nosocomial UTIs (41). These outcomes differ from a study in Peshawar, Pakistan that documented the prevalence of *E. coli* isolates at 12% and 21% from pus and urine cultures, respectively, whereas none of *E. coli* strains were isolated from blood cultures (42). These differences in the prevalence of *E. coli* isolates from urine and pus cultures could be associated with variations in sample size, demographic characteristics, and methodology of research or perhaps virulence of locally carried strains.

The emergence of MDR *E. coli* isolates is causing therapeutic failures that are a serious public health threat and lead to high morbidities and mortalities in hospital settings (43). This study reported a high resistance rate for fluoroquinolones (95–100%), cephalosporins (88–95%), penicillins (82–97%), and aminoglycosides (58%). These findings are in agreement with the recent studies reporting increasing AMR in the South Asian region (44, 45). A recent study analyzed the AMR rates for GLASS specified pathogen/antimicrobials combination from Pakistan (2006–2018) and reported a high resistance rate (>50%) to fluoroquinolones, 3rd generation cephalosporins among the *E. coli* and *K. pneumoniae* (46). The variation in results may be due to the variation in quality and standardization of antimicrobial sensitivity testing methods used in hospitals.

Our study exhibited the prevalence of multiple $bla_{NDM}$ variants such as $bla_{NDM-1}$, $bla_{NDM-5}$, $bla_{NDM-20}$, and $bla_{NDM-21}$, among clinical *E. coli* isolates. These findings are similar to other studies that indicate the widespread distribution of carbapenemases genes globally (47–50). In this study, NDM-5 was observed to be most prevalent among the carbapenem-resistant *E. coli* isolates, similar to other studies published in Pakistan and China (33, 51). Since the isolation of $bla_{NDM-5}$ in Henan in 2013, its detection rate is continuously increasing, and now it has emerged as a prominent subtype of $bla_{NDM}$ (52). Several factors responsible for the dissemination of $bla_{NDM}$-carrying isolates include irrational use of broad-spectrum antimicrobials, self-medication, easy availability of antimicrobials at pharmacies without the prescription of doctors, and substandard

antimicrobials. All these factors are also considered a prominent source of the horizontal spread of superbugs among individuals (53).

To have better knowledge regarding the virulence potential of isolates, it is necessary to know the occurrence of virulence genes among these isolates. The most frequently occurring virulence genes among carbapenem-resistant *E. coli* isolates in this study are *terC, traT, gad, sitA, fyuA, irp2, CapU,* and *iss*. These findings differ from the studies conducted in Iran and Egypt that reported the presence of *traT, fimH,iutA, csgA, hlyA,* and *crl* genes more frequently in carbapenem-resistant ExPEC isolates and revealed that UPEC isolates were more pathogenic than others (54). The prevalence of *traT* gene contributes to the serum resistance of isolates as they become able to avoid complement systems, which increases the risk of causing septic shock and the rate of mortality (55). Pathogenicity of *E. coli* isolates was further examined by establishing the *G. mellonella* model experiment. We observed dose dependent response on larval survival, as the survival of larvae decreases by increasing the inocula of *E. coli* strains. In this study, most isolates with the exception of PK-5081, PK-5127, PK-5138, and PK-5178 showed negligible pathogenicity. This is likely due to the fact that many of the *E. coli* strains carried in Pakistan are of low virulence potential with only a few strains possessing additional virulence factors such as *iucC, iutA, sat, kpsE, kpsMII_K5,* and *papA_F43* found in more pathogenic strains listed above. Another study used the *G. mellonella* model to examine the pathogenicity of ExPEC isolates and presented a notable correlation between the virulence potential of isolates and virulence gene repertoire. The higher number of virulence genes in ExPEC isolates was responsible for the rapid death of the larvae (56).

We found diverse *E. coli* STs, the most predominant was ST-405 followed by ST-167 and ST-361. Our findings are similar to another study conducted in Pakistan that reported ST405 in *E. coli* isolates (51). The most pathogenic isolate of NDM-producing *E. coli* belonging to ST405 is most commonly present in Asia and other regions of the world (57–60). ST167 NDM-producing *E. coli* strains are causing infections worldwide (61, 62), which created great interest and attention. Notably, *E. coli* strains carrying the $bla_{NDM}$ gene belonging to ST167 have been reported in companion animals (63, 64), which suggests the transmission of ST167 *E. coli* harboring the $bla_{NDM-5}$ gene between humans and animals. Notably, we did not find any *E. coli* strain positive for the $bla_{NDM}$ gene associated with ST131 in the current study.

The plasmid replicon typing analysis revealed different replicon types including IncF, IncI1, IncI2, IncX3, and IncY. IncF replicon type was most common in this study with sub-replicons such as IncFIA, IncFII, IncFIC, and IncFIB. In previous studies, different plasmid types harboring the $bla_{NDM}$ gene reported include IncFIA, IncFIB (65), IncHI1, IncFIIA, and IncN (66), IncX3 (67), IncB/O (68), IncFIC, IncF, and IncK (69), and IncY, IncA/C, and IncI1 (70). Plasmids can acquire different resistance genes or transposons and are responsible for the spread of high levels of AMR (8). Out of 34 isolates, 10 harbored IncI1 plasmid, which belongs to the narrow range of host plasmid type and was only observed in Enterobacterales. Several studies have suggested that IncI1 plasmids predominantly carry genes encoding for AMR, particularly for the ESBLs genes (71, 72).

## Conclusion

Carbapenem-resistance has been considered as one of the most significant menaces to global healthcare, and the prevalence of NDM variants in clinical *E. coli* isolates has further increased this threat. Therefore, early detection of the $bla_{NDM}$ possessing *E. coli* isolates with any decreased sensitivity to the carbapenems is crucial for choosing the most appropriate antibiotic therapy and applying additional efficient infection control measures. The limited use of antibiotics, particularly carbapenems and cephalosporins, may help to prevent the emergence of such resistance-patterns. Furthermore, robust and comprehensive infection control management in the hospital is required to avoid such infections.

## AUTHOR AFFILIATIONS

[1]Institute of Microbiology, University of Agriculture, Faisalabad, Pakistan
[2]Department of Medical Microbiology, School of Medicine, Institute of Infection and Immunity, Cardiff University, Cardiff, United Kingdom
[3]Biomedical Research Center, Qatar University, Doha, Qatar

## AUTHOR ORCIDs

Mark A. Toleman  http://orcid.org/0000-0002-9497-0512

## FUNDING

| Funder | Grant(s) | Author(s) |
| --- | --- | --- |
| International Research Support Initiative Scheme, Higher Education Commission (HEC), Pakistan | Project No. 1-8/HEC/HRD/ 2021/10873 | Sabahat Abdullah |

## AUTHOR CONTRIBUTIONS

Sabahat Abdullah, Formal analysis, Investigation, Methodology, Visualization, Writing – original draft | Abdulrahman Almusallam, Investigation, Methodology | Mei Li, Investigation, Methodology | Muhammad Shahid Mahmood, Investigation, Methodology | Nahla O. Eltai, Investigation, Methodology | Mark A. Toleman, Conceptualization, Formal analysis, Investigation, Methodology, Resources, Software, Supervision, Writing – original draft, Writing – review and editing | Mashkoor Mohsin, Conceptualization, Data curation, Formal analysis, Funding acquisition, Investigation, Methodology, Project administration, Resources, Software, Supervision, Validation, Visualization, Writing – original draft, Writing – review and editing.

## DATA AVAILABILITY

The genomes of *E. coli* isolates have been submitted to the NCBI BioProject database under accession number PRJNA932156.

## ETHICS APPROVAL

This study was approved by the Institutional Biosafety Committee (IBC) D. No. 8025/ORIC of University of Agriculture, Faisalabad.

## ADDITIONAL FILES

The following material is available online.

### Supplemental Material

**Supplemental Table S1 (Spectrum00584-23-S0001.docx).** Antimicrobial susceptibility.
**Supplemental Table S2 (Spectrum00584-23-S0002.docx).** Isolate descriptions.
**Supplemental Table S3 (Spectrum00584-23-S0003.xlsx).** Number of SNPs exhibited between each isolate.

### Open Peer Review

**PEER REVIEW HISTORY (review-history.pdf).** An accounting of the reviewer comments and feedback.

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
