## [Reviewer comments · Microbiology Spectrum]

Microbiology Spectrum

Whole genome-based genetic insights of bla_{NDM} producing clinical *E. coli* isolates in hospitals settings of Pakistan

Sabahat Abdullah, Abdul Rahman Almusallam, Mei Li, Muhammad Mahmood, Muhammad Mushtaq, Nahla Eltai, Mark Toleman, and Mashkoor Mohsin

Corresponding Author(s): Mark Toleman, Cardiff University, and Mashkoor Mohsin, University of Agriculture Faisalabad

Review Timeline:

Submission Date:	February 8, 2023
Editorial Decision:	April 22, 2023
Revision Received:	June 13, 2023
Accepted:	July 2, 2023

Editor: Sadjia Bekal

Reviewer(s): The reviewers have opted to remain anonymous.

Transaction Report:

DOI: <https://doi.org/10.1128/spectrum.00584-23>

April 22, 2023

Dr. Mark Alexander Toleman
Cardiff University
Infection and Immunity Cardiff, School of Medicine United Kingdom
Cardiff CF144XN
United Kingdom

Re: Spectrum00584-23 (Whole genome-based genetic insights of bla_{NDM} producing clinical *E. coli* isolates in hospitals settings of Pakistan)

Dear Dr. Mark Alexander Toleman:

Link Not Available

Sincerely,

Sadjia Bekal

Journals Department
Reviewer comments:

Reviewer #1 (Comments for the Author):

The study presented herein is intriguing as it provides insights into the genomic environment of clinical *E. coli* carrying bla_{NDM} in Pakistan. The article reported on the isolation and antimicrobial susceptibility testing of 34 bla_{NDM}-carrying *E. coli* strains, as well as genome-wide analysis. Although the authors have made valuable contributions to the understanding of the genetic environment of NDM-carrying strains, a few comments are necessary to improve the manuscript's quality:

Line 43: The manuscript contains several spelling errors, including the misspelling of 'a-' as 'a' in some instances. To ensure the manuscript's accuracy and precision, these errors should be corrected.

Line 177-178: Accession numbers of sequenced data generated in this study should be updated if possible.

Line 214: It is recommended to correct the spelling of 'b-lactamases' to ' β -lactamases' in the manuscript.

Line 240: 'n=01' needs to be corrected to 'n=1'.

Line 245: To provide a more comprehensive understanding of the phylogenetic relationships among the strains, additional information regarding the tree-building method is necessary. Specifically, the manuscript should include details on the sequencing data used for the analysis, such as whether SNP or core genome-based methods were employed, the length of the core genome used, the software and methodology utilized for the tree construction (such as ML or NJ), and if appropriate outgroups were used.

Line 246: '7' needs to be corrected to 'n=7'.

Line 248: While the current phylogenetic tree suggests that ST167 may not belong to a single cluster and that SNP distances between clusters are significant, it is possible that alternative tree-building methods could result in different conclusions. It may be necessary for the authors to explore additional tree-building methods beyond what is presented in the manuscript. Additionally, it is recommended to use the same color for ST1702 in Figure 1.

Line 273: Figures 2A, 2B, and 2D appear to have been stretched and distorted, and the sizes of the figures are not consistent with one another. The authors should adjust the figures to ensure that they are accurately and clearly presented.

Line 294: The bleMBL gene should be labeled on Figure 3.

Line 295: Although there are some differences in the gene structures of Groups C and D, Figure 3 shows that most of the regions of the two groups share a high degree of identity. The authors may need to provide a detailed explanation to address this apparent discrepancy.

Line 298: The genetic environment of blaNDM gene located in the IncHI2 plasmid pPK-5160-blaNDM-1 was not fully described and discussed in this study.

Reviewer #2 (Comments for the Author):

In the present manuscript entitled 'Whole genome-based genetic insights of blaNDM producing clinical E. coli isolates from 2 hospitals in Pakistan, the authors raised an important topic: the importance to characterise the genetic environment of antimicrobial resistance genes especially those involved in the resistance of last-resort antibiotics such as carbapenems. The authors concentrated their work on the main prevalent carbapenemase gene in India, blaNDM.

They showed that distinct plasmid replicons are involved in the spread of these genes.

Methods

The methods used are well described. However, the results of AST would have provided important information.

Interpretation

The two hospitals being located in the same area of Pakistan, Faisalabad, can the blaNDM distribution described be representative of the one of Pakistan?

Are some of these strains part of outbreaks?

In the abstract (lane 36), the authors wrote that 'virulence correlated with the number of virulence determinants' while in lanes 307-308 and lanes 355-356 they conclude that 'most isolates... showed negligible pathogenicity. What is the real impact of such virulence factors on the pathogenicity of the strains studied?

Results

Table 1 - Could it be added as supplemental figure?

Table 2 - Did the authors described the characteristics of the blaNDM-positive plasmids from 10 representative E. coli ST? If so, would it be pertaining to include the genes actually detected on the described plasmid i.e. PK-5160 seem to carry blaNDM-1 on the only IncHI2 plasmid.

A table including antimicrobial susceptibility data should be added.

Table 3 - The words 'E. coli strains' are missing in the title.

The authors should better explain the similarity of the 4 groups of IncF plasmids described figure 3. Such a similarity look like a signature of genetic exchange leading to the mosaic plasmids described.

Major modifications

The authors performed AST. It could have been relevant to correlate the presence of the predicted AMR genes described in the paper to the antimicrobial susceptibility testing results. As stated in the conclusion, 'Early detection of the blaNDM ...with decreased sensitivity to carbapenems is crucial'... excepted that the 12 antimicrobials tested are not mentioned (lines 98:100),

neither their MIC result provided.

Staff Comments:

Preparing Revision Guidelines

Please return the manuscript within 60 days; if you cannot complete the modification within this time period, please contact me. If you do not wish to modify the manuscript and prefer to submit it to another journal, please notify me of your decision immediately so that the manuscript may be formally withdrawn from consideration by Microbiology Spectrum.

Manuscript Title: Whole genome-based genetic insights of *bla*_{NDM} producing clinical *E. coli* isolates in hospitals settings of Pakistan

Summary

In the present manuscript entitled 'Whole genome-based genetic insights of *bla*_{NDM} producing clinical *E. coli* isolates from 2 hospitals in Pakistan, the authors raised an important topic: the importance to characterise the genetic environment of antimicrobial resistance genes especially those involved in the resistance of last-resort antibiotics such as carbapenems. The authors concentrated their work on the main prevalent carbapenemase gene in India, *bla*_{NDM}.

They showed that distinct plasmid replicons are involved in the spread of these genes.

Methods

The methods used are well described. However, the results of AST would have provided important information.

Interpretation

The two hospitals being located in the same area of Pakistan, Faisalabad, can the *bla*_{NDM} distribution described be representative of the one of Pakistan?

Are some of these strains part of outbreaks?

In the abstract (lane 36), the authors wrote that 'virulence correlated with the number of virulence determinants' while in lanes 307-308 and lanes 355-356 they conclude that 'most isolates... showed negligible pathogenicity. What is the real impact of such virulence factors on the pathogenicity of the strains studied?

Results

Table 1 - Could it be added as supplemental figure?

Table 2 - Did the authors described the characteristics of the *bla*_{NDM}-positive plasmids from 10 representative *E. coli* ST? If so, would it be pertaining to include the genes actually detected on the described plasmid i.e. PK-5160 seem to carry *bla*_{NDM-1} on the only IncHI2 plasmid.

A table including antimicrobial susceptibility data should be added.

Table 3 - The words '*E. coli* strains' are missing in the title.

The authors should better explain the similarity of the 4 groups of IncF plasmids described figure 3. Such a similarity look like a signature of genetic exchange leading to the mosaic plasmids described.

Major modifications

The authors performed AST. It could have been relevant to correlate the presence of the predicted AMR genes described in the paper to the antimicrobial susceptibility testing results. As stated in the conclusion, 'Early detection of the *bla*_{NDM} ...with decreased sensitivity to carbapenems is crucial'... excepted that the 12 antimicrobials tested are not mentioned (lines 98:100), neither their MIC result provided.

Response to Reviewers

Reviewer #1 (Comments for the Author):

The study presented herein is intriguing as it provides insights into the genomic environment of clinical *E. coli* carrying blaNDM in Pakistan. The article reported on the isolation and antimicrobial susceptibility testing of 34 blaNDM-carrying *E. coli* strains, as well as genome-wide analysis. Although the authors have made valuable contributions to the understanding of the genetic environment of NDM-carrying strains, a few comments are necessary to improve the manuscript's quality:

Line 43: The manuscript contains several spelling errors, including the misspelling of 'a-' as 'a' in some instances. To ensure the manuscript's accuracy and precision, these errors should be corrected.

Response: Lines 42 to 44. Morbidity and mortality caused by multidrug-resistant (MDR) bacteria are increasing globally, with a recent study estimating the global burden of AMR at 4.95 million deaths in 2019 (2).

Corrections have been made throughout the manuscript.

Line 177-178: Accession numbers of sequenced data generated in this study should be updated if possible.

Response: Lines 173 to 174. The genomes of *E. coli* isolates have been submitted to NCBI BioProject database (<https://www.ncbi.nlm.nih.gov/bioproject/>) under the bioproject PRJNA932156.

Line 214: It is recommended to correct the spelling of 'b-lactamases' to ' β -lactamases' in the manuscript.

Response: Spellings of β -lactamases have been corrected throughout the manuscript.

Line 240: 'n=01' needs to be corrected to 'n=1'.

Response: Line 259-261. The phylogenetic tree showed that among 34 distinct *E. coli* strains, 2 belong to unknown STs, and 32 belong to 8 ST types: ST405 (n=11), ST167 (n=7), ST361 (7), ST156 (n=2), ST2851 (n=2), ST1702 (n=2), and ST2450 (n=1).

Correction has been made.

Line 245: To provide a more comprehensive understanding of the phylogenetic relationships among the strains, additional information regarding the tree-building method is necessary. Specifically, the manuscript should include details on the sequencing data used for the analysis, such as whether SNP or core genome-based methods were employed, the length of the core genome used, the software and methodology utilized for the tree construction (such as ML or NJ), and if appropriate outgroups were used.

Response: Lines 240-258. Phylogenetic relationships among *E. coli* isolates were determined by using the online tool CSI Phylogeny (1.4 version) (<https://cge.cbs.dtu.dk/services/CSIPhylogeny/>). The pipeline comprises of various freely available programs. The paired-end reads from each isolate were aligned against the reference genome, using Burrows-Wheeler Aligner (BWA), SAMtools, “mpileup” command and bedtools. The single nucleotide polymorphism (SNP) based phylogenetic tree was generated by calling and filtering SNPs, site validation and phylogeny based on a concatenated alignment of the high-quality SNPs. For inferring phylogeny, the analysis was run with the standard parameters and the NCTC11129 strain genome (GenBank accession number NZ_LR134222.1) was used as a reference sequence. The analysis was run with the default parameters; a minimal depth at SNP positions was ten reads along with a relative depth at SNP positions of 10%, a minimal distance between SNPs was of 10 bp, a minimal SNPs quality was 30 and a minimal Z-score was of 1.96, a minimal SNP quality was 30 and a minimal read mapping quality was of 25. The Z-score expresses the confidence with which a base was called at a given position and the Z-score was 1.96. Number of SNPs exhibited among closely related isolates were calculated using distance matrix file generated as a result of phylogeny (35). The output core alignment file was used to construct the Maximum-likelihood tree with 1000 bootstrap replications, by using MEGA-X v 10.0.5 (<https://www.megasoftware.net/home>) (36). The phylogenetic tree of the alignments was visualized and edited by iTOL v 4.4.2 software (<https://itol.embl.de>) (37).

Line 246: ' 7' needs to be corrected to 'n=7'.

Response: Lines 259-261. The phylogenetic tree showed that among 34 distinct *E. coli* strains, 2 belong to unknown STs, and 32 belong to 8 ST types: ST405 (n=11), ST167 (n=7), ST361 (n=7), ST156 (n=2), ST2851 (n=2), ST1702 (n=2), and ST2450 (n=1).

Correction has been made.

Line 248: While the current phylogenetic tree suggests that ST167 may not belong to a single cluster and that SNP distances between clusters are significant, it is possible that alternative tree-building methods could result in different conclusions. It may be necessary for the authors to explore additional tree-building methods beyond what is presented in the manuscript. Additionally, it is recommended to use the same color for ST1702 in Figure 1.

Response: Maximum likelihood phylogenetic tree was constructed using core genome SNPS. (Figure 3).

Lines 262-263. Notably, the reference genome grouped with three isolates belonging to ST156 and unknown ST, has been added in manuscript.

Color correction for ST1702 has been made.

Line 273: Figures 2A, 2B, and 2D appear to have been stretched and distorted, and the sizes of the figures are not consistent with one another. The authors should adjust the figures to ensure that they are accurately and clearly presented.

Response: Figure: Sizes of figures have been adjusted properly for clear presentation. (Figure 2)

Line 294: The *ble*_{MBL} gene should be labeled on Figure 3.

Response: The *ble*_{MBL} gene has been labeled in Figure 3.

Line 295: Although there are some differences in the gene structures of Groups C and D, Figure

3 shows that most of the regions of the two groups share a high degree of identity. The authors may need to provide a detailed explanation to address this apparent discrepancy.

Response: Lines 303-316. The genetic environment around the *bla*_{NDM} gene located on IncF plasmids can be classified into three types of groups. These regions carrying by *bla*_{NDM} gene were surrounded by IS26. For group A (*IS26-ΔISAbal25-bla*_{NDM}-*ble*_{MBL}-*trpF-ISCRI-sul1-qacE-aadA2-dfrA12-IntI1-IS26-ΔTnAS1*) the flanking genetic structure of *bla*_{NDM} gene was composed of an IS26 and incomplete ISAbal25 interrupted by insertion sequence ISCRI and the genes *ble*_{MBL} (bleomycin resistance), *trpF* (phosphoribosylanthranilate isomerase), *sul1* (sulfonamide-resistant dihydropteroate synthase), *qacE* (quaternary ammonium compound resistance protein), *aadA2* (aminoglycoside nucleotidyltransferase), *dfrA12* (dihydrofolate reductase), and *IntI1* (class 1 integron integrase) located downstream. Furthermore, this group also has downstream addition of *ΔTnAS1* associated with IS26. Compared with group A, group B (*IS26-IS26-ΔISAbal25-bla*_{NDM}-*ble*_{MBL}-*trpF-ISCRI-sul1-qacE-aadA2-dfrA12-IntI1-IS26-ΔTnAS1*) includes another IS26 upstream addition and completed downstream deletion of *ΔTnAS1*. Group C (*IS26-ΔISAbal25-bla*_{NDM}-*ble*_{MBL}-*trpF-ISCRI-sul1-qacE-aadA2-dfrA12-IntI1-ΔTnAS3-IS26*) includes the downstream addition of *ΔTnAS3* associated with oppositely directed IS26 (Figure 3).

A detailed explanation has been added in the manuscript. Figure 3 has also been updated.

Line 298: The genetic environment of *bla*_{NDM} gene located in the IncHI2 plasmid pPK-5160-*bla*_{NDM-1} was not fully described and discussed in this study.

Response: Lines 317-320. One IncHI2 type *bla*_{NDM-1} bearing plasmid pPK-5160-*bla*_{NDM-1} was also identified in PK-5160 strain. The core genetic environment of *bla*_{NDM-1} gene in this plasmid contains multiple antibiotic resistance genes including *tetR*, *tetA*, *bla*_{NDM-1}, and *ble*_{MBL} surrounded by IS26 and IS3000 upstream and downstream from *bla*_{NDM-1} (Figure 4).

Added in manuscript.

Reviewer #2 (Comments for the Author):

In the present manuscript entitled 'Whole genome-based genetic insights of blaNDM producing clinical E. coli isolates from 2 hospitals in Pakistan, the authors raised an important topic: the importance to characterise the genetic environment of antimicrobial resistance genes especially those involved in the resistance of last-resort antibiotics such as carbapenems. The authors concentrated their work on the main prevalent carbapenemase gene in India, blaNDM. They showed that distinct plasmid replicons are involved in the spread of these genes.

Methods

The methods used are well described. However, the results of AST would have provided important information.

Interpretation

The two hospitals being located in the same area of Pakistan, Faisalabad, can the blaNDM distribution described be representative of the one of Pakistan?

Response: Faisalabad is the third largest city of Pakistan, and second largest city of wider Punjab region. Allied and DHQ, two tertiary care hospitals in this city, offer a variety of healthcare facilities to a large number of populations of the city and other distant rural areas. This city could serve as a representative of large population of Pakistan.

Are some of these strains part of outbreaks?

Response: This study collected E. coli strains from laboratories of two hospitals, isolated from urine and pus cultures of in and outpatients. These strains were not part of outbreaks.

In the abstract (lane 36), the authors wrote that 'virulence correlated with the number of virulence determinants' while in lanes 307-308 and lanes 355-356 they conclude that 'most isolates... showed negligible pathogenicity. What is the real impact of such virulence factors on the pathogenicity of the strains studied?

Response: The investigation of virulence factors on the pathogenicity of studied strains may facilitate the novel therapeutics, the improvement of design of effective vaccines, and the prevention of further spreading of the multidrug resistant isolates.

Results

Table 1 - Could it be added as supplemental figure?

Response: This table has been added as a supplementary table 2.

Table 2 - Did the authors described the characteristics of the bla_{NDM}-positive plasmids from 10 representative E. coli ST? If so, would it be pertaining to include the genes actually detected on the described plasmid i.e. PK-5160 seem to carry bla_{NDM}-1 on the only IncHI2 plasmid.

Response: Authors selected the isolates for comparative analysis of plasmids on the basis of Inc types of plasmids as well as bla_{NDM} genes harbored by these isolates. Isolate PK-5160 was selected for analysis as it harbored bla_{NDM}-1 gene and IncHI2 type of different Inc types of plasmids.

A table including antimicrobial susceptibility data should be added.

Response: A supplemental table 1 showing the results of antimicrobial susceptibility data has been added.

Table 3 - The words 'E. coli strains' are missing in the title.

Response: **Table 2. Characteristics of *E. coli* strains harboring bla_{NDM} gene**

The words 'E. Coli strains' has been added in title of table 2.

The authors should better explain the similarity of the 4 groups of IncF plasmids described figure 3. Such a similarity look like a signature of genetic exchange leading to the mosaic plasmids described.

Response: Lines 303-316. The genetic environment around the bla_{NDM} gene located on IncF plasmids can be classified into three types of groups. These regions carrying by bla_{NDM} gene were surrounded by IS26. For group A (*IS26-ΔISAbal25-bla_{NDM}-ble_{MBL}-trpF-ISCRI-sull-qacE-aadA2-dfrA12-IntI1-IS26-ΔTnAS1*) the flanking genetic structure of bla_{NDM} gene was composed

of an *IS26* and incomplete *ISAbal25* interrupted by insertion sequence *ISCR1* and the genes *ble_{MBL}* (bleomycin resistance), *trpF* (phosphoribosylanthranilate isomerase), *sulI* (sulfonamide-resistant dihydropteroate synthase), *qacE* (quaternary ammonium compound resistance protein), *aadA2* (aminoglycoside nucleotidyltransferase), *dfrA12* (dihydrofolate reductase), and *IntI1* (class 1 integron integrase) located downstream. Furthermore, this group also has downstream addition of $\Delta TnAS1$ associated with *IS26*. Compared with group A, group B (*IS26-IS26- Δ ISAbal25-*bla_{NDM}*-*ble_{MBL}*-*trpF*-*ISCR1*-*sulI*-*qacE*-*aadA2*-*dfrA12*-*IntI1*-*IS26*- $\Delta TnAS1$*) includes another *IS26* upstream addition and completed downstream deletion of $\Delta TnAS1$. Group C (*IS26- Δ ISAbal25-*bla_{NDM}*-*ble_{MBL}*-*trpF*-*ISCR1*-*sulI*-*qacE*-*aadA2*-*dfrA12*-*IntI1*- $\Delta TnAS3$ -*IS26**) includes the downstream addition of $\Delta TnAS3$ associated with oppositely directed *IS26* (Figure 3).

A detailed explanation has been added in the manuscript. Figure 3 has also been updated.

Major modifications

The authors performed AST. It could have been relevant to correlate the presence of the predicted AMR genes described in the paper to the antimicrobial susceptibility testing results. As stated in the conclusion, 'Early detection of the *bla_{NDM}* ...with decreased sensitivity to carbapenems is crucial'... excepted that the 12 antimicrobials tested are not mentioned (lines 98:100), neither their MIC result provided.

Response:

A table showing the results of antibiotic susceptibility testing has been added as a supplemental table 1.

Antimicrobial susceptibility testing revealed that all 34 *E. coli* isolates were MDR strains, and they were resistant to multiple categories of antibiotics (n>3) (Table S1 in supplemental material). Therefore, each isolate carried at least three categories of resistance genes associated with resistance phenotype (Table 1). Almost all these isolates were non-susceptible to fluoroquinolones (95-100%), cephalosporins (87-95%) and penicillins (82-97%). Moreover, over half of the isolates were resistant to aminoglycosides (57%). The MICs results of CR-EC isolates revealed 100% susceptibility to tigecycline and colistin.

Has also been added in manuscript.

July 2, 2023

Dr. Mark Alexander Toleman
Cardiff University
Infection and Immunity Cardiff, School of Medicine United Kingdom
Cardiff CF144XN
United Kingdom

Re: Spectrum00584-23R1 (Whole genome-based genetic insights of bla_{NDM} producing clinical *E. coli* isolates in hospitals settings of Pakistan)

Dear Dr. Mark Alexander Toleman:

Your manuscript has been accepted, and I am forwarding it to the ASM Journals Department for publication. You will be notified when your proofs are ready to be viewed.

Sincerely,

Sadjia Bekal
Editor, Microbiology Spectrum
